# Generalized Derangetropy Functionals for Modeling Cyclical Information Flow

**DOI:** 10.3390/e27060608

**Published:** 2025-06-07

**Authors:** Masoud Ataei, Xiaogang Wang

**Affiliations:** 1Department of Mathematical and Computational Sciences, University of Toronto, Mississauga, ON L5L 1C6, Canada; 2Department of Mathematics and Statistics, York University, Toronto, ON M3J 1P3, Canada; stevenw@yorku.ca

**Keywords:** information dynamics, probability distributions, functional information theory, entropy, nonlinear differential equations, cyclical information flow

## Abstract

This paper introduces a functional framework for modeling cyclical and feedback-driven information flow using a generalized family of derangetropy operators. In contrast to scalar entropy measures such as Shannon entropy, these operators act directly on probability densities, providing a topographical representation of information across the support of the distribution. The proposed framework captures periodic and self-referential aspects of information evolution through functional transformations governed by nonlinear differential equations. When applied recursively, these operators induce a spectral diffusion process governed by the heat equation, with convergence toward a Gaussian characteristic function. This convergence result establishes an analytical foundation for describing the long-term dynamics of information under cyclic modulation. The framework thus offers new tools for analyzing the temporal evolution of information in systems characterized by periodic structure, stochastic feedback, and delayed interaction, with potential applications in artificial neural networks, communication theory, and non-equilibrium statistical mechanics.

## 1. Introduction

The propagation of information in complex systems frequently exhibits temporally structured and spatially modulated patterns, shaped by feedback, recurrence, and periodic forcing. Such dynamics are pervasive in biological networks, cognitive systems, and cyclic physical processes, where probability distributions evolve under local stochastic interactions subject to global structural constraints. Classical entropy measures, particularly Shannon entropy [1], have long served as foundational tools for quantifying uncertainty. However, their scalar and static formulation renders them inadequate for describing the evolving structure of informational states in systems governed by feedback or periodic modulation. This limitation is particularly evident in contexts such as neural computation or intracellular signaling, where distributions evolve through temporally coordinated interactions and self-regulating mechanisms.

To address these limitations, the present work develops a generalized framework for derangetropy, a recently introduced class of entropy-modulated functional operators designed to model cyclical and feedback-driven information flow [2]. Originally proposed as a localized operator acting on probability densities, derangetropy departs from traditional scalar entropy metrics by defining density-like functionals that reflect cumulative distributional structure and internal modulation. This formulation enables the characterization of systems in which informational complexity emerges through interactions between local stochasticity and global structural constraints.

Building on this foundation, we introduce and analyze three distinct types of derangetropy operators, each capturing a different mode of probabilistic transformation. Type-I derangetropy induces an entropy-attenuating operator that sharpens low-entropy regions while preserving the support of the original distribution. This behavior is characteristic of systems exhibiting structural persistence under dynamical accumulation, such as resting-state cortical activity. In contrast, Type-II derangetropy acts as an entropy-amplifying operator, enhancing high-entropy regions and capturing dispersive phenomena typical of turbulent or unstable environments.

Type-III derangetropy departs from entropy-coupled modulation and instead introduces a phase-sensitive transformation governed by a quadratic sine function. This class is particularly suited to modeling resonance-driven dynamics observed in oscillatory systems, including electrophysiological and mechanical oscillators. Notably, the recursive application of derangetropy operators induces a universal diffusion behavior governed by the heat equation in the spectral domain, with convergence toward a Gaussian characteristic function. This convergence result provides a unifying analytical perspective across the derangetropy classes and offers a tractable framework for analyzing long-term informational dynamics in modulated or cyclic systems.

The generalized derangetropy operators proposed herein enable localized redistribution of probability mass through structural and dynamical modulation. By encoding entropy coupling, periodicity, and feedback into self-referential functionals, this framework extends classical entropy theory into new domains. The associated convergence result further connects local transformations to global dynamical outcomes, offering insight into the stabilization, amplification, or oscillation of informational structure across time and space.

The remainder of the paper is organized as follows. Section 2 reviews prior work on information dynamics across scientific domains. Section 3 defines Type-I derangetropy and analyzes its governing differential equation and self-modulating properties. Section 4 introduces Type-II derangetropy and characterizes its entropy-amplifying transformation. Section 5 develops Type-III derangetropy and presents a convergence analysis emphasizing Gaussianization and spectral diffusion. We conclude in Section 7 with a discussion of future directions, including multivariate extensions and potential applications in structured probabilistic modeling and interacting dynamical systems.

## 2. Related Work

Recent efforts to characterize information flow in systems with cyclical, delayed, or feedback-driven dynamics span multiple domains. Foundational work in inference theory has reinterpreted entropy as an operator for updating probability distributions rather than merely quantifying uncertainty [3]. This shift unifies principles of maximum entropy and Bayesian updating, framing information flow as a structured transformation of distributions in response to incoming evidence or internal modulation.

In complex biological systems, particularly biochemical and regulatory networks, fluctuations in molecular states interact with energetic constraints to shape signal fidelity and processing efficiency [4,5]. Empirical analyses suggest that such systems balance specificity and noise suppression by modulating structural interactions over time. These dynamics call for tools that capture not only uncertainty but also its redistribution across support.

Spectrally resolved methods have also emerged to study how information propagates over multiple timescales. Frequency-based metrics have been shown to influence predictive capacity and memory formation in stochastic systems [6], with parallels in biological regulation where noise shaping and gating enable adaptive signal control [7]. These developments underscore the growing relevance of spectral analysis in models of time-evolving uncertainty.

Generalized entropies have been proposed to address limitations of the Shannon framework in high-dimensional, non-Gaussian environments. Notably, the extension of the Mutual Information Matrix (MIM) to Rényi entropies of an arbitrary order introduces a spectral perspective for quantifying dependencies [8]. The structure of these matrices reveals how informational couplings shift under different entropy orders, highlighting the influence of rare or nonlinear events in complex dynamical systems.

In neuroscience, periodic synchronization underlies functional connectivity and information routing. Cortical oscillations reflect hierarchical coordination mechanisms that require models incorporating temporal structure and feedback sensitivity [9]. Similar principles arise in engineered systems, such as communication channels affected by cyclostationary noise, where capacity varies with periodic modulation [10,11]. These phenomena motivate the integration of feedback and time-sensitive structure in information-theoretic modeling.

Control theory and machine learning further illustrate the need for temporally aware formulations. Recurrent feedback mechanisms give rise to non-Markovian dynamics, challenging classical tools based on mutual or directed information [12,13]. New approaches are required to track distributional evolution across iterative feedback loops, especially when state transitions are governed by complex, delayed interactions.

Thermodynamic studies of open systems have linked information metrics such as mutual information and entropy production to subsystem coupling [14,15]. In quantum regimes, coherence introduces additional layers of complexity in how uncertainty and information are structured and exchanged [16]. Recent insights in stochastic control have shown that time-modulated feedback and noise can shape the temporal profile of information transfer, yielding dynamic patterns not easily captured by static metrics [17].

Network-theoretic perspectives also emphasize structural constraints in information propagation. Statistical physics models have revealed how topological features, such as modularity and degree distribution, affect entropy flow across nodes and links [18]. These insights suggest that heterogeneous architectures require formulations that adapt to localized asymmetries in information transmission.

In ecological and financial systems, adaptive information flow responds to environmental volatility. Studies have shown that temporal plasticity, structural feedback, and bidirectional causality, as in the relationship between realized and implied volatilities in markets, require metrics capable of capturing dynamic interdependence [19,20].

Lastly, the behavior of entropy estimators under finite samples has been a focus of statistical analysis. Bias and variance corrections for Shannon entropy estimators are necessary to avoid misleading inferences, especially in highly skewed or undersampled distributions [21]. This body of work highlights the importance of structure-aware techniques that maintain robustness across practical data limitations.

## 3. Type-I Derangetropy

### 3.1. Definition

Consider a probability space (Ω,F,P), where Ω denotes the sample space, F is a σ-algebra of measurable subsets of Ω, and P is a probability measure defined on F. Let X:Ω→R be a real-valued random variable that is measurable with respect to the Borel σ-algebra $B_{R}$ on R. Suppose that *X* has an absolutely continuous distribution with an associated probability density function (PDF) f∈$L^{2}\left(R,B_{R},\lambda \right)$, where λ represents the Lebesgue measure on R. The cumulative distribution function (CDF) corresponding to *X* is given by$$F\left(x\right)=\int_{-\infty }^{x}f\left(t\right)\;d\lambda \left(t\right),\;x\in R.$$

We now introduce the notion of Type-I derangetropy, which refines probability distributions while incorporating entropy-based modulation.

**Definition** **1**(Type-I Derangetropy)**.**
*The Type-I derangetropy functional $\rho :L^{2}\left(R,B_{R},\lambda \right)\to L^{2}\left(R,B_{R},\lambda \right)$ is defined by the following mapping*(1)$$\rho \left[f\right]\left(x\right)=\left(\frac{24}{\pi e}\right)\sin\left(\pi F\left(x\right)\right)F{\left(x\right)}^{F(x)}{\left(1-F\left(x\right)\right)}^{1-F(x)}f\left(x\right).$$
*Alternatively, it can be rewritten as a Fourier-type transformation*
(2)$$\rho \left[f\right]\left(x\right)=\left(\frac{24}{\pi e}\right)\sin\left(\pi F\left(x\right)\right)\;e^{-H_{B}\left(F\left(x\right)\right)}f\left(x\right),$$
*where*
(3)$$H_{B}\left(F\left(x\right)\right)=-F\left(x\right)\log\left(F\left(x\right)\right)-\left(1-F\left(x\right)\right)\log\left(1-F\left(x\right)\right),$$
*is the Shannon entropy for a Bernoulli distribution with success probability p=F(x). The evaluation of the derangetropy functional of Type-I at a specific point x∈R is denoted by $\rho_{f}\left(x\right)$.*

The sine function in the transformation plays a crucial role in systems that exhibit periodicity, where information alternates between concentrated and dispersed states. This function serves as both a modulation mechanism and a projection operator. As a modulation mechanism, it introduces controlled oscillations that reflect inherent periodic characteristics in the probability distribution. The oscillatory nature of the sine function ensures that the transformation redistributes probability mass while preserving essential features of the original density. As a projection operator, it maps probability densities into a transformed space where local variations become more prominent, allowing for a more refined probabilistic representation. This behavior is particularly relevant in cyclic systems such as time-series processes and wave-based phenomena, where periodic effects shape the behavior of probability distributions.

The presence of $H_{B}\left(F\left(x\right)\right)$ in the transformation introduces a localized uncertainty measure, quantifying how probability mass is distributed at any given point. This entropy function describes the informational balance between the cumulative probability F(x) and its complement 1−F(x), providing a measure of relative uncertainty in the distribution. By incorporating entropy directly into the transformation, Type-I derangetropy adapts the probability density in a manner influenced by local uncertainty. The weighting factor $e^{-H_{B}\left(F\left(x\right)\right)}$ modulates the extent to which the transformation affects the distribution, attenuating its influence in high-entropy regions while allowing stronger reshaping in low-entropy areas. This adaptive adjustment ensures that the transformation refines the distribution without enforcing uniform smoothing, preserving key probabilistic characteristics.

The explicit presence of f(x) in the transformation ensures that Type-I derangetropy refines probability densities rather than reconstructing them entirely. Unlike convolution-based smoothing techniques or regularization approaches, this transformation maintains the local structural integrity of the distribution while selectively modifying probability mass based on entropy considerations. In regions of low entropy, the transformation retains the original density, preventing unnecessary redistribution. In contrast, in high-entropy regions, where uncertainty is greater, it modulates probability values in a controlled manner, refining the density without excessive distortion. This property distinguishes Type-I derangetropy from conventional entropy-based transformations that often impose global constraints or flatten distributions.

Type-I derangetropy is best suited for systems that exhibit a combination of periodicity and self-regulation, where information accumulates in a structured yet dynamic fashion. A canonical example lies in biological neural systems, such as resting-state electroencephalography (EEG) or functional magnetic resonance imaging (fMRI) signals, where regions of the brain alternate between high and low informational states. In such systems, the entropy-attenuating nature of Type-I ensures that stable, low-entropy zones retain their shape while more uncertain, transitional regions are selectively reshaped. This allows for the extraction of latent functional hierarchies that are otherwise obscured by global entropy measures like Shannon or Rényi entropy. In practice, when applied to empirical distributions over space or time, Type-I derangetropy acts as a local contrast enhancer, refining meaningful variations without flattening them, ideal for denoising or revealing subtle periodic signatures in oscillatory yet organized domains, such as circadian gene expressions, ecological cycles, or cortical oscillations.

Moreover, while the definition above assumes an absolutely continuous distribution with respect to the Lebesgue measure, the formulation of Type-I derangetropy naturally extends to discrete distributions. In such cases, the CDF F(x) becomes a step function constructed from the probability mass function (PMF), and the derangetropy functional is interpreted over the discrete support. The structural form of the transformation remains valid; i.e.,(4)$$\rho_{p}\left(x\right)=\left(\frac{24}{\pi e}\right)\sin\left(\pi F\left(x\right)\right)e^{-H_{B}\left(F\left(x\right)\right)}p\left(x\right),$$
where p(x)=P(X=x) is the PMF of a discrete random variable *X*. This discrete analogue preserves the entropy-modulated, localized refinement properties of the continuous version, and is particularly suited to systems governed by discrete probabilistic events such as count processes or stochastic automata.

### 3.2. Mathematical Properties

We now establish key mathematical properties of the Type-I derangetropy functional. For any absolutely continuous PDF f(x), the functional ρ[f](x) is a nonlinear operator that belongs to the space $C^{\infty }\left(R\right)$ and remains a valid PDF.

**Theorem** **1.**
*For any absolutely continuous f(x), the derangetropy functional $\rho_{f}\left(x\right)$ is a valid probability density function.*


**Proof.** To prove that $\rho_{f}\left(x\right)$ is a valid PDF, we need to show that $\rho_{f}\left(x\right)\geq 0$ for all x∈R and that $\int_{-\infty }^{\infty }\rho_{f}\left(x\right)dx=1$. The non-negativity of $\rho_{f}\left(x\right)$ is clear due to the non-negativity of its components. To verify the normalization condition, a change of variable z=F(x) is applied. Since dz=f(x)dx, this transformation allows the integral of $\rho_{f}\left(x\right)$ to be rewritten in terms of *z* as follows:(5)$$\int_{-\infty }^{\infty }\rho_{f}\left(x\right)dx=\int_{0}^{1}\left(\frac{24}{\pi e}\right)\sin\left(\pi z\right)z^{z}{\left(1-z\right)}^{1-z}dz.$$ As established in [2], the integral is(6)$$\int_{0}^{1}\sin\left(\pi z\right)z^{z}{\left(1-z\right)}^{1-z}dz=\frac{\pi e}{24},$$
which, in turn, implies that(7)$$\int_{-\infty }^{\infty }\rho_{f}\left(x\right)dx=1.$$ Hence, $\rho_{f}\left(x\right)$ is a valid PDF. □

Since $\rho_{f}\left(x\right)$ is a valid PDF, it encapsulates the informational content of the original distribution f(x) while recursively uncovering a hierarchical structure of its own information content. This iterative refinement is governed by the following recurrence relation(8)$$\rho_{f}^{\left(n\right)}\left(x\right)=\left(\frac{24}{\pi e}\right)\sin\left(\pi G_{f}^{\left(n-1\right)}\left(x\right)\right)e^{-H_{B}\left(G_{f}^{\left(n-1\right)}\left(x\right)\right)}\rho_{f}^{\left(n-1\right)}\left(x\right),$$
where $\rho_{f}^{\left(n\right)}\left(x\right)$ represents the *n*th iteration of the derangetropy functional, and$$G_{f}^{\left(n\right)}\left(x\right)=\int_{-\infty }^{x}\rho_{f}^{\left(n\right)}\left(t\right)dt$$
is the associated CDF. The initial conditions for this recursion are given by$$\rho_{f}^{\left(0\right)}\left(x\right)=f\left(x\right)\;\mathrm{and}\;G_{f}^{\left(0\right)}\left(x\right)=F\left(x\right),$$
where f(x) and F(x) denote the PDF and CDF of the original distribution, respectively. This recursive structure ensures that each subsequent iteration $\rho_{f}^{\left(n\right)}\left(x\right)$ integrates the informational content of all preceding layers, thereby constructing a hierarchical probabilistic representation of information.

Moreover, the self-referential hierarchical structure distinguishes Type-I derangetropy from conventional entropy-based transformations, which typically impose global constraints or flatten distributions. Furthermore, since the transformation retains absolute continuity, the process maintains smoothness across iterations, allowing for a deeper probabilistic analysis over successive refinements.

The Type-I derangetropy functional exhibits a structured refinement process, systematically redistributing probability mass while preserving fundamental probabilistic properties. This refinement is characterized by a second-order nonlinear differential equation that governs the evolution of probability accumulation in distribution functions. The interaction between entropy-based modulation and probabilistic redistribution ensures that the transformation dynamically adjusts the density function while retaining key informational characteristics.

An illustrative case emerges when considering a uniform distribution over the interval (0,1), where the PDF is given by f(x)=1 and the corresponding CDF is F(x)=x. Under these conditions, the governing differential equation encapsulates an intricate equilibrium between entropy suppression, logistic accumulation, and oscillatory refinement.

**Theorem** **2.**
*Let X be a random variable following a uniform distribution on the interval (0, 1). Then, the Type-I derangetropy functional $\rho_{f}\left(x\right)$ satisfies the following second-order nonlinear ordinary differential equation*


(9)
$$\frac{d^{2}\rho_{f}\left(x\right)}{dF{\left(x\right)}^{2}}+2\log\left(\frac{1-F(x)}{F(x)}\right)\frac{d\rho_{f}\left(x\right)}{dF(x)}+\left[\pi^{2}-\frac{1}{F(x)(1-F(x))}+\log^{2}\left(\frac{1-F(x)}{F(x)}\right)\right]\rho_{f}\left(x\right)=0,$$

*where the initial conditions are set as*

$${\left.\rho_{f}\left(x\right)\right|}_{F(x)=0}=0,\;\;\mathrm{and}\;{\left.\;\frac{d\rho_{f}\left(x\right)}{dF(x)}\right|}_{F(x)=0}=\frac{24}{e}.$$



**Proof.** To solve this differential equation, we introduce an integrating factor μ(F(x)) to eliminate the first-order derivative term. As established in [2], this integrating factor is given by(10)$$\mu \left(F\left(x\right)\right)=e^{F(x)\log(F(x))+(1-F(x))\log(1-F(x))}.$$ By rewriting the exponential term, this expression simplifies to(11)$$\mu \left(F\left(x\right)\right)=F{\left(x\right)}^{F(x)}{\left(1-F\left(x\right)\right)}^{1-F(x)}.$$ Multiplying the entire equation by this integrating factor transforms it into a simpler form, allowing us to define a new function u(F(x)), such that(12)$$\rho_{f}\left(x\right)=\mu \left(F\left(x\right)\right)u\left(F\left(x\right)\right).$$ Substituting this transformation into the differential equation, the first-order derivative term cancels, reducing the equation to(13)$$\frac{d^{2}u\left(F\left(x\right)\right)}{dF{\left(x\right)}^{2}}+\pi^{2}u\left(F\left(x\right)\right)=0.$$ This equation is a standard second-order homogeneous linear differential equation with constant coefficients. Its general solution is(14)$$u\left(F\left(x\right)\right)=C_{1}\sin\left(\pi F\left(x\right)\right)+C_{2}\cos\left(\pi F\left(x\right)\right),$$
where $C_{1}$ and $C_{2}$ are constants. Finally, evaluating the initial values yields $C_{1}=\frac{24}{\pi e}$ and $C_{2}=0$, implying that $\rho_{f}\left(x\right)$ is indeed a solution to the differential equation. □

The differential equation governing Type-I derangetropy describes how probability density evolves under its transformation, revealing a deep connection between probability accumulation and entropy modulation. The presence of the log-odds function, $\log\left(\frac{1-F(x)}{F(x)}\right)$, naturally arises in entropy-driven transformations and provides insight into the underlying refinement process. Since this function encodes the relative probability mass distribution, its role in the differential equation suggests that the rate of change in probability density is governed by the balance of mass accumulation within the total probability space. The transformation redistributes probability mass in a way that maintains equilibrium between entropy modulation and structured probabilistic refinement, ensuring that high-uncertainty regions are dynamically adjusted while preserving the essential probabilistic features of the original distribution.

The derivative of the log-odds function, given by(15)$$\frac{d}{dF(x)}\log\left(\frac{1-F(x)}{F(x)}\right)=\frac{d}{dF(x)}H_{B}\left(F\left(x\right)\right)=\frac{1}{F(x)(1-F(x))}$$
quantifies the local entropy gradient, which dictates the strength of the refinement process. The transformation is most active near F(x)=0.5, where entropy gradients reach their maximum, and is progressively weaker as F(x)→0 or F(x)→1, where entropy naturally diminishes. This implies that the transformation does not merely smooth probability distributions but rather introduces structured refinements that reflect localized entropy variations. In other words, the transformation modulates probability mass according to the entropy landscape, amplifying changes where uncertainty is highest and stabilizing regions where probability mass is more concentrated. The function $\log\left(\frac{1-F(x)}{F(x)}\right)$ is anti-symmetric about F(x)=0.5, indicating that the governing equation differentiates probability mass accumulation on either side of the median. This property ensures that refinements occur in a balanced manner, preventing distortions in probability redistribution. That is, the structured modulation introduced by Type-I derangetropy ensures that probability adjustments remain coherent across the entire distribution, avoiding artificial drifts that would compromise the probabilistic integrity of the refined density function.

The presence of the term $\pi^{2}$ in the differential equation highlights the intrinsic oscillatory nature of the refinement process. This term ensures that the transformation does not merely act as a smoothing operator but instead introduces a structured modulation mechanism that preserves critical features of the original distribution. The oscillatory behavior dictated by this term prevents excessive entropy flattening while allowing the transformation to refine probability densities dynamically. As a result, the governing equation encodes an evolving probability transformation where entropy regulation, probabilistic redistribution, and oscillatory modulation contribute to a structured refinement process that maintains coherence in probability mass adjustments.

## 4. Type-II Derangetropy

### 4.1. Definition

The Type-II derangetropy functional represents a refinement process that enhances uncertainty while maintaining the overall probabilistic structure of the distribution. Unlike its Type-I counterpart, which redistributes probability mass in a manner that mitigates uncertainty, Type-II derangetropy operates through an entropy-amplifying transformation that reinforces the contribution of high-entropy regions. The notion of the Type-II derangetropy functional is formally defined below.

**Definition** **2**(Type-II Derangetropy)**.**
*The Type-II derangetropy functional $\tau :L^{2}\left(R,B_{R},\lambda \right)\to L^{2}\left(R,B_{R},\lambda \right)$ is defined by the following mapping*(16)$$\tau \left[f\right]\left(x\right)=\left(\frac{e}{\pi }\right)\frac{\sin(\pi F(x))}{F{\left(x\right)}^{F(x)}{\left(1-F\left(x\right)\right)}^{1-F(x)}}f\left(x\right),$$
*or, alternatively, as a Fourier-type transformation*
(17)$$\tau \left[f\right]\left(x\right)=\left(\frac{e}{\pi }\right)\sin\left(\pi F\left(x\right)\right)\;e^{H_{B}\left(F\left(x\right)\right)}f\left(x\right).$$
*The evaluation of the derangetropy functional of Type-II at a specific point x∈R is denoted by $\tau_{f}\left(x\right)$.*

The defining characteristic of Type-II derangetropy is its ability to amplify entropy through the weighting factor $e^{H_{B}\left(F\left(x\right)\right)}$, which enhances probability mass in high-entropy regions while reducing its relative influence in areas of low entropy. Unlike Type-I derangetropy, which suppresses entropy to structure probability redistribution, Type-II derangetropy increases entropy by emphasizing uncertainty-driven refinements. Rather than redistributing probability mass uniformly, the transformation reshapes the density in a way that increases the contribution of regions where entropy is higher while reducing the dominance of more concentrated areas.

The sinusoidal term sin(πF(x)) ensures that the transformation remains structured by regulating how entropy amplification modifies probability mass. Without this modulation, the entropy-amplified density $e^{H_{B}\left(F\left(x\right)\right)}f\left(x\right)$ would introduce large variations, potentially distorting the overall density. The sine function interacts with entropy amplification to control the redistribution process, reinforcing probability density adjustments while preventing excessive deviation from the original distribution. This interaction ensures that refinements follow a controlled transformation rather than an unbounded expansion of probability mass.

Unlike Type-I derangetropy, which refines probability density while constraining uncertainty, Type-II derangetropy amplifies entropy while directing the redistribution of probability mass according to its entropy profile. This does not merely shift probability mass to high-entropy regions but increases their relative contribution while reducing the influence of low-entropy areas. The transformation does not lead to indiscriminate dispersion but instead enforces an entropy-driven redistribution that aligns with the intrinsic uncertainty of the distribution.

The self-regulating nature of Type-II derangetropy ensures that entropy amplification follows an adaptive process where probability mass is redistributed in accordance with entropy gradients. Instead of increasing entropy arbitrarily, the transformation adjusts the probability density function in a way that preserves its structural properties while refining its information content. Unlike standard transformations that impose smoothing effects or regularization constraints, Type-II derangetropy enhances entropy while preserving meaningful distinctions within the probability distribution. This controlled entropy amplification suggests applications in probability density refinement, entropy-aware transformations, and statistical modeling, where increasing entropy while maintaining structural coherence is crucial for probabilistic inference.

Type-II derangetropy becomes relevant in contexts where high-entropy regions are not noise to be suppressed but rather signals of inherent system dynamics, such as in stochastic processes, turbulent flows, or volatile financial markets. By amplifying entropy instead of attenuating it, Type-II derangetropy captures the natural tendency of these systems to drift into disordered or diffusive states. For instance, in the modeling of high-frequency trading behavior or biophysical systems under random forcing e.g., ion channel kinetics and intracellular transport, local entropy peaks are often indicative of regime transitions or instability fronts. Type-II derangetropy allows these zones to be magnified, helping to identify precursors of phase change or collapse. Additionally, its application is beneficial in epidemiological modeling of contagion, where regions of maximal uncertainty, e.g., transmission rate variability, are critical to the propagation mechanism and thus should be weighted more heavily in information-theoretic assessments. In essence, Type-II is a tool for systems where informational disorder is not just present, but mechanistically central.

### 4.2. Empirical Observations

The behavior of Type-I and Type-II derangetropy functionals highlights fundamental differences in probability mass refinement. Each functional encodes a distinct form of entropy modulation, shaping probability density according to structured probability evolution and entropy-aware transformations. Figure 1 examines five representative distributions—uniform, normal, exponential, semicircle, and arcsine—illustrating how these functionals respond to symmetry, skewness, and boundary effects.

For the uniform distribution, where probability is evenly distributed across its support, Type-I derangetropy reshapes it into a semi-parabolic density, peaking at the median where entropy is most balanced. As probability mass moves toward the boundaries, entropy decreases, leading to a decline in functional values. The Type-II transformation further enhances central probability concentration, resulting in a bell-shaped density that decays more sharply near the edges.

For the normal distribution, both transformations preserve symmetry while refining probability mass around the mean. Type-I derangetropy applies a smooth redistribution, maintaining the general shape, while Type-II amplifies the central peak, increasing probability concentration in high-entropy regions and compressing lower-probability areas.

For the exponential distribution, which exhibits strong skewness, both functionals adjust the density while retaining its asymmetry. Type-I derangetropy smooths out the distribution, moderating the steep decline at low values while preserving the overall shape. Type-II intensifies probability concentration in the high-density region, further emphasizing entropy-driven refinements while reducing influence in lower-probability areas.

For the semicircle distribution, both functionals preserve its compact support and symmetry but differ in their refinement strategies. Type-I maintains the overall shape while introducing mild adjustments near the median. Type-II amplifies probability mass at the center, reducing density near the boundaries, and reflecting its tendency to reinforce entropy-dominant regions more aggressively.

For the arcsine distribution, where probability mass is highly concentrated at the boundaries, the transformations yield particularly distinct refinements. Type-I derangetropy redistributes mass away from the edges, producing an inverted semi-parabolic density that spreads probability more evenly. Type-II generates a semi-parabolic shape peaking at the median, significantly reducing mass at the boundaries and reinforcing entropy-maximizing regions. This contrast underscores the functional distinction: while Type-I ensures proportional redistribution, Type-II concentrates mass in high-entropy areas, diminishing lower-entropy contributions.

These observations demonstrate that Type-I derangetropy functions as an entropy-balancing transformation, refining probability distributions while preserving structural proportions. Type-II derangetropy, in contrast, acts as an entropy-enhancing transformation, intensifying probability mass concentration in high-entropy regions while suppressing low-entropy contributions. This distinction makes Type-II particularly suited for applications requiring stronger probability centralization, while Type-I is more appropriate for cases demanding balanced entropy refinement.

### 4.3. Mathematical Properties

We now establish the key mathematical properties of the Type-II derangetropy functional that underscore its utility in analyzing probability distributions. For any absolutely continuous PDF f(x), the derangetropy functional τ[f](x) is a nonlinear operator that belongs to the space $C^{\infty }\left(R\right)$ having the following first derivative (18)$$\frac{d}{dx}\tau_{f}\left(x\right)=\tau_{f}\left(x\right)\left[\pi \cot\left(\pi F\left(x\right)\right)+\log\left(\frac{F(x)}{1-F(x)}\right)+\frac{f^{\prime }\left(x\right)}{f(x)}\right],$$
where the derivatives are taken with respect to *x*.

The following theorem shows that the Type-II derangetropy maps the PDF f(x) into another valid probability density function.

**Theorem** **3.**
*For any absolutely continuous f(x), the Type-II derangetropy functional $\tau_{f}\left(x\right)$ is a valid probability density function.*


**Proof.** To prove that $\tau_{f}\left(x\right)$ is a valid PDF, we need to show that $\tau_{f}\left(x\right)\geq 0$ for all x∈R and that $\int_{-\infty }^{\infty }\tau_{f}\left(x\right)\;dx=1$. The non-negativity of $\tau_{f}\left(x\right)$ is clear due to the non-negativity of the terms involved in its definition. The normalization condition can further be verified by the change of variables z=F(x), yielding(19)$$\int_{-\infty }^{\infty }\tau_{f}\left(x\right)\;dx=\int_{0}^{1}\left(\frac{e}{\pi }\right)\frac{\sin(\pi z)}{z^{z}{\left(1-z\right)}^{1-z}}dz.$$ As shown in Appendix A, the integral(20)$$\int_{0}^{1}\frac{\sin(\pi z)}{z^{z}{\left(1-z\right)}^{1-z}}dz=\frac{\pi }{e},$$
which, in turn, implies that(21)$$\int_{-\infty }^{\infty }\tau_{f}\left(x\right)\;dx=1.$$ Hence, $\tau_{f}\left(x\right)$ is a valid PDF. □

As a valid PDF, the Type-II derangetropy functional $\tau_{f}\left(x\right)$ enables the formulation of a self-referential framework, where the informational content of each successive layer is recursively expressed in terms of previous layers. This hierarchical structure captures how probability mass evolves under repeated applications of Type-II derangetropy, encoding both entropy amplification and oscillatory refinement.

The recursive formulation follows from the following transformation(22)$$\tau_{f}^{\left(n\right)}\left(x\right)=\left(\frac{e}{\pi }\right)\sin\left(\pi G_{f}^{\left(n-1\right)}\left(x\right)\right)e^{H_{B}\left(G_{f}^{\left(n-1\right)}\left(x\right)\right)}\tau_{f}^{\left(n-1\right)}\left(x\right),$$
where $\tau_{f}^{\left(n\right)}\left(x\right)$ represents the *n*th iteration of the transformation, and$$G_{f}^{\left(n\right)}\left(x\right)=\int_{-\infty }^{x}\tau_{f}^{\left(n\right)}\left(t\right)dt$$
is the CDF at the *n*th stage. The recursion is initialized by setting$$\tau_{f}^{\left(0\right)}\left(x\right)=f\left(x\right)\;\mathrm{and}\;G_{f}^{\left(0\right)}\left(x\right)=F\left(x\right).$$

This formulation ensures that each subsequent layer $\tau_{f}^{\left(n\right)}\left(x\right)$ encapsulates the cumulative effect of all preceding transformations. The iterative refinement process progressively modulates probability redistribution based on the evolving entropy landscape, reinforcing high-entropy regions while systematically adjusting lower-entropy areas. This recursive structure highlights the fundamental role of Type-II derangetropy as an adaptive entropy-enhancing transformation, allowing probability distributions to evolve through a controlled sequence of refinements.

The Type-II derangetropy functional further satisfies a nonlinear ordinary differential equation given below.

**Theorem** **4.**
*Let X be a random variable following a uniform distribution on the interval (0, 1). Then, the derangetropy functional $\tau_{f}\left(x\right)$ satisfies the following second-order nonlinear ordinary differential equation*

(23)
$$\frac{d^{2}\tau_{f}\left(x\right)}{dF{\left(x\right)}^{2}}-2\log\left(\frac{1-F(x)}{F(x)}\right)\frac{d\tau_{f}\left(x\right)}{dF(x)}+\left[\pi^{2}+\frac{1}{F(x)(1-F(x))}+\log^{2}\left(\frac{1-F(x)}{F(x)}\right)\right]\tau_{f}\left(x\right)=0,$$

*where the initial conditions are set as*

$${\left.\tau_{f}\left(x\right)\right|}_{F(x)=0}=0,\;\;\mathrm{and}\;{\left.\;\frac{d\tau_{f}\left(x\right)}{dF(x)}\right|}_{F(x)=0}=e.$$



**Proof.** To solve the differential equation, we introduce an integrating factor μ(F(x)) to eliminate the first-order derivative term, which is given by(24)$$\mu \left(F\left(x\right)\right)=e^{F\left(x\right)\log\left(\frac{1-F(x)}{F(x)}\right)}{\left(1-F\left(x\right)\right)}^{-1}.$$ Rewriting the exponential term, this simplifies to(25)$$\mu \left(F\left(x\right)\right)=F{\left(x\right)}^{-F(x)}{\left(1-F\left(x\right)\right)}^{-(1-F(x))}.$$ Multiplying the entire equation by this integrating factor transforms it into a simpler form, allowing us to define a new function u(F(x)), such that(26)$$\tau_{f}\left(x\right)=\mu \left(F\left(x\right)\right)u\left(F\left(x\right)\right).$$ Substituting this transformation into the differential equation, the first-order derivative term cancels, reducing the equation to(27)$$\frac{d^{2}u\left(F\left(x\right)\right)}{dF{\left(x\right)}^{2}}+\pi^{2}u\left(F\left(x\right)\right)=0.$$ This equation is a second-order homogeneous linear differential equation with constant coefficients, whose general solution is(28)$$u\left(F\left(x\right)\right)=C_{1}\sin\left(\pi F\left(x\right)\right)+C_{2}\cos\left(\pi F\left(x\right)\right),$$
where $C_{1}$ and $C_{2}$ are constants. To determine these constants, we apply the initial conditions, which yields $C_{1}=\frac{e}{\pi }$ and $C_{2}=0$, implying that $\tau_{f}\left(x\right)$ is indeed a solution to the differential equation. □

The governing differential equations for Type-I and Type-II derangetropy reveal a fundamental mathematical duality, illustrating how both transformations share the same underlying structure while diverging in their treatment of entropy gradients and probability evolution. This structural connection is encoded in the opposing signs of key terms, specifically, the first-order derivative coefficient and the entropy-dependent potential, which dictate distinct modes of probability refinement.

The log-odds function, appearing in both equations, serves as the fundamental driver of probability redistribution, quantifying the relative balance of probability mass on either side of a given point. In Type-I derangetropy, the positive coefficient of the first derivative term ensures that entropy gradients modulate probability refinement in a balanced manner, preventing excessive concentration while redistributing probability mass proportionally across the domain. The negative entropy correction term reinforces this structured refinement, ensuring that mass is neither excessively centralized nor overdispersed.

The differential equation for Type-II derangetropy introduces an inverse transformation by reversing the sign of the first derivative term, altering the interaction between probability mass and entropy gradients. This reversal amplifies the influence of high-entropy regions, strengthening probability concentration where uncertainty is greatest while reducing mass in low-entropy regions. The positive entropy correction term further reinforces this effect, ensuring that the transformation enhances contrast in probability distribution, rather than merely diffusing mass. Unlike Type-I derangetropy, which ensures entropy balancing across the domain, Type-II derangetropy enforces entropy amplification, strengthening dominant probability regions and reducing the presence of structured low-entropy areas.

This duality between the governing differential equations highlights the complementary nature of Type-I and Type-II derangetropy as entropically conjugate processes. Under Type-I derangetropy, the evolution of probability mass follows a structured refinement, ensuring entropy-controlled redistribution without excessive dominance of high-entropy regions. In contrast, under Type-II derangetropy, probability mass evolution reinforces entropy amplification, emphasizing refinements that concentrate probability mass in regions of highest uncertainty while suppressing low-entropy influence. The governing equations formalize this interaction between structured refinement and entropy-enhancing transformations, demonstrating that while distinct in their mechanisms, these transformations together define a unified framework for entropy-aware probability evolution.

## 5. Type-III Derangetropy

### 5.1. Definition

The Type-III derangetropy functional introduces a non-entropy-based transformation of probability densities, designed to capture oscillatory and resonance-governed redistribution mechanisms. Unlike its Type-I and Type-II counterparts, which hinge on entropy attenuation and amplification, Type-III leverages phase modulation, using the CDF as a proxy for normalized phase.

**Definition** **3**(Type-III Derangetropy)**.**
*The Type-III derangetropy functional $\nu :L^{2}\left(R,B_{R},\lambda \right)\to L^{2}\left(R,B_{R},\lambda \right)$ is defined by the following mapping*(29)$$\nu_{f}\left(x\right)=2\sin^{2}\left(\pi F\left(x\right)\right)\;f\left(x\right).$$
*The evaluation of the derangetropy functional of Type-III at a specific point at x∈R is denoted by $\nu_{f}\left(x\right)$.*

This transformation constitutes a deterministic, nonlinear modulation of the input density, with the kernel $2\sin^{2}\left(\pi F\left(x\right)\right)$ introducing a wave-like amplification that peaks at F(x)=0.5 and vanishes at the boundaries of the distribution support (F(x)=0 or 1). It enacts a phase-locked redistribution of mass toward central probability regions and attenuates the tails, yielding a shaped refinement that reflects an intrinsic balance across the support.

The phase-modulation mechanism implemented by the quadratic sine function encodes a symmetry consistent with harmonic structures: periodic, smooth, and centered. Unlike entropy-driven methods, which reshape the distribution in response to localized uncertainty, Type-III derangetropy is insensitive to local entropy gradients. Instead, it processes the distribution globally via its CDF, effectively normalizing the support and embedding the density into a circular or toroidal geometry. This embedding allows the transformation to capture cyclic symmetry, which would be obscured in linear coordinate representations.

From an operational standpoint, Type-III can be viewed as a probabilistic analog of bandpass filtering in signal processing, concentrating density in regions of spectral equilibrium while suppressing outliers. Importantly, the modulation factor is bounded within [0,2], ensuring that the transformed density remains normalizable and analytically tractable under repeated application. This boundedness distinguishes Type-III from sharpening kernels which may become numerically unstable or degenerate under iteration.

Mathematically, the repeated application of the Type-III operator induces smoothing in the Fourier domain. This behavior arises because the sinusoidal modulation in the probability domain corresponds to a convolution-like contraction in the characteristic function, thereby acting as a low-pass filter. Over successive iterations, this spectral contraction drives the distribution toward a Gaussian profile, a fact established in the proceeding convergence results. This process mirrors spectral diffusion observed in thermodynamically driven stochastic systems and establishes a bridge between phase-modulated density transformations and heat kernel smoothing on the probability simplex.

The applications of Type-III derangetropy span several domains where phase alignment and wave-based coherence dominate over mere entropy accumulation. In neuroscience, for example, the transformation is well-suited for extracting rhythmic activity across theta, alpha, beta, and gamma bands in EEG signals, where spectral localization and symmetry are paramount. In such contexts, classical entropy metrics may obfuscate phase-dominant features due to their bias toward flattening. Type-III instead enhances structured oscillatory components, maintaining alignment with periodic temporal behavior.

In engineered systems such as radar and sonar, Type-III can be used to accentuate return signals with coherent phase trajectories, effectively serving as a probabilistic phase-regularization operator. In optical physics, particularly in laser and interferometric analysis, the ability to redistribute energy across a normalized phase spectrum is analogous to cavity-mode shaping or phase-front correction. Likewise, in financial econometrics, where cyclical volatility patterns emerge, e.g., through intraday cycles, this phase-modulated transformation may help characterize regime-dependent behaviors without enforcing entropy-based distortions.

Conceptually, Type-III derangetropy emphasizes the role of global structure over local randomness. It encodes an operational belief that in certain classes of systems, particularly those governed by cyclic, resonant, or harmonic principles, information propagates not via entropy gradients but through structured, phase-aware reconfigurations. By eschewing direct entropy dependence, this transformation offers a unique lens for examining density evolution where preservation of symmetry, periodicity, or resonance is prioritized.

Hence, Type-III derangetropy presents a theoretically bounded, spectrally coherent, and phase-sensitive functional for probability refinement. Its cumulative phase-based modulation aligns naturally with applications involving resonance, coherence, and cyclic symmetry, offering an alternative to entropy-centric transformations in both the analysis and modeling of complex systems.

### 5.2. Mathematical Properties

For any absolutely continuous PDF f(x), the Type-III derangetropy functional ν[f](x) is a nonlinear operator that belongs to the space $C^{\infty }\left(R\right)$ having the following first derivative (30)$$\frac{d}{dx}\nu_{f}\left(x\right)=\nu_{f}\left(x\right)\left[2\pi \cot\left(\pi F\left(x\right)\right)f\left(x\right)+\frac{f^{\prime }\left(x\right)}{f(x)}\right],$$
where the derivatives are taken with respect to *x*.

The following theorem shows that the Type-III derangetropy maps the PDF f(x) into another valid probability density function.

**Theorem** **5.**
*For any absolutely continuous f(x), the Type-III derangetropy functional $\nu_{f}\left(x\right)$ is a valid probability density function.*


**Proof.** To prove that $\nu_{f}\left(x\right)$ is a valid PDF, we need to show that $\nu_{f}\left(x\right)\geq 0$ for all x∈R and that $\int_{-\infty }^{\infty }\nu_{f}\left(x\right)\;dx=1$. The non-negativity of $\nu_{f}\left(x\right)$ is clear due to the non-negativity of the terms involved in its definition. The normalization condition can further be verified by the change of variables z=F(x), yielding(31)$$\int_{-\infty }^{\infty }\nu_{f}\left(x\right)\;dx=\int_{0}^{1}2\sin^{2}\left(\pi z\right)\;dz=\int_{0}^{1}\left(1-\cos\left(2\pi z\right)\right)dz=1.$$ Hence, $\nu_{f}\left(x\right)$ is a valid PDF. □

Similar to its Type-I and Type-II counterparts, the Type-III derangetropy functional $\nu_{f}\left(x\right)$ has self-referential properties, which are expressed as follows:(32)$$\nu_{f}^{\left(n\right)}\left(x\right)=2\sin^{2}\left(\pi G_{f}^{\left(n-1\right)}\left(x\right)\right)\nu_{f}^{\left(n-1\right)}\left(x\right),$$
where $\nu_{f}^{\left(n\right)}\left(x\right)$ represents the *n*th iteration of the transformation, and$$G_{f}^{\left(n\right)}\left(x\right)=\int_{-\infty }^{x}\nu_{f}^{\left(n\right)}\left(t\right)dt$$
is the CDF at the *n*th stage. The initial conditions for the recursion are set as$$\nu_{f}^{\left(0\right)}\left(x\right)=f\left(x\right)\;\mathrm{and}\;G_{f}^{\left(0\right)}\left(x\right)=F\left(x\right).$$

The recursive formulation of Type-III derangetropy highlights a structured refinement process where each successive iteration enhances the probabilistic modulation applied in the previous step. Unlike Type-I, which balances entropy gradients, and Type-II, which amplifies entropy regions, Type-III enforces a periodic redistribution mechanism governed by the sine-squared function. This transformation progressively sharpens probability mass near mid-range values of the CDF while suppressing contributions from extreme regions. As a result, with each iteration, the density function becomes increasingly concentrated in regions where $\sin^{2}\left(\pi F\left(x\right)\right)$ is maximized, leading to a self-reinforcing oscillatory refinement.

Figure 2 illustrates how successive iterations of Type-III derangetropy transform different probability distributions. Despite the absence of explicit entropy terms, the iterative application of Type-III derangetropy exhibits a clear pattern of redistribution that progressively refines probability mass. The transformation does not merely preserve the shape of the original density but instead pushes probability toward a more structured, centralized form. Regardless of the initial distribution, the repeated application of Type-III derangetropy leads to a progressively smoother and more bell-shaped structure, ultimately converging toward a normal distribution.

For the uniform distribution, the initial transformation reshapes the density into a symmetric bell-like function, emphasizing probability concentration near the median. The second iteration intensifies this effect, reducing mass at the edges and further reinforcing the central region. A similar pattern is observed in the normal distribution, where each iteration enhances probability mass concentration near the mean while reducing the density of the tails. The effect is more apparent in skewed distributions such as the exponential distribution, where the first transformation shifts probability mass away from the heavy tail, while the second iteration further reduces the density gradient, making the distribution increasingly symmetric.

This process is evident in compact distributions such as the semicircle, where each transformation reduces the impact of boundary effects, favoring a more centralized redistribution of probability mass. In the case of the arcsine distribution, which is initially highly concentrated at the boundaries, the transformation effectively mitigates this extreme localization, redistributing probability toward the interior and gradually shifting toward a bell-shaped distribution. The common trend across all distributions is that Type-III derangetropy iteratively smooths and reshapes probability densities in a structured manner, reinforcing central regions while gradually eliminating extremities. The transformation acts as a refinement mechanism that reinforces a well-defined probabilistic structure, ensuring that the density evolves in a direction consistent with the Gaussian limit.

The differential equation governing the Type-III derangetropy functional for a uniform distribution, as stated in the following theorem, reveals fundamental principles of wave-based probability refinement and structured probability evolution.

**Theorem** **6.**
*Let X be a random variable following a uniform distribution on the interval (0,1). Then, the derangetropy functional $\nu_{f}\left(x\right)$ satisfies the following third-order ordinary differential equation*

(33)
$$\frac{d^{3}\nu_{f}\left(x\right)}{dF{\left(x\right)}^{3}}+4\pi^{2}\frac{d\nu_{f}\left(x\right)}{dF(x)}=0,$$

*where the initial conditions are set as*

$${\left.\nu_{f}\left(x\right)\right|}_{F(x)=0}=0,\;{\left.\;\frac{d\nu_{f}\left(x\right)}{dF(x)}\right|}_{F(x)=0}=0\;\;\mathrm{and}\;{\left.\;\frac{d^{2}\nu_{f}\left(x\right)}{dF{\left(x\right)}^{2}}\right|}_{F(x)=0}=4\pi^{2}.$$



**Proof.** The first and third derivatives of $\nu_{f}\left(x\right)$ are computed as follows:(34)$$\frac{d\nu_{f}\left(x\right)}{dF(x)}=2\pi \sin\left(2\pi F\left(x\right)\right),$$
and(35)$$\frac{d^{3}\nu_{f}\left(x\right)}{dF{\left(x\right)}^{3}}=-8\pi^{3}\sin\left(2\pi F\left(x\right)\right),$$
respectively. Substituting these expressions into the differential equation, we obtain(36)$$-8\pi^{3}\sin\left(2\pi F\left(x\right)\right)+4\pi^{2}\left(2\pi \sin\left(2\pi F\left(x\right)\right)\right)=0,$$
which implies that $\nu_{f}\left(x\right)$ satisfies the given third-order differential equation. □

The governing equation for Type-III derangetropy is a third-order linear differential equation with the following characteristic equation(37)$$r^{3}+4\pi^{2}r=0,$$
whose roots are as follows:(38)$$r_{1}=0,\;\mathrm{and}\;r_{2},r_{3}=\pm 2\pi i.$$
This indicates a combination of stationary and oscillatory modes, implying that the probability refinement process follows a wave-like evolution rather than simple diffusion. Unlike purely diffusive models, which allow probability mass to spread indefinitely, the presence of oscillatory solutions implies a cyclic probability refinement mechanism, where mass is systematically redistributed in a structured manner.

The third-order nature of this equation introduces higher-order control over probability redistribution, capturing not only the rate of change of probability mass but also its jerk (the rate of change of acceleration). This distinguishes Type-III derangetropy from simple diffusion-based models, which typically involve only first- or second-order derivatives. The third derivative term ensures dynamic feedback regulation, where probability mass alternates between expansion and contraction, maintaining a structured oscillatory refinement process.

The term $4\pi^{2}\frac{d\nu_{f}\left(x\right)}{dF(x)}$ plays a crucial role in reinforcing oscillatory stabilization. Unlike a conventional damping mechanism, which suppresses oscillations over time, this term acts as a restorative force, ensuring that probability oscillations remain well-structured while preventing uncontrolled dispersion. This prevents the transformation from excessively smoothing out the density, instead enforcing a controlled wave-based refinement of probability mass. This structured evolution explains why Type-III derangetropy systematically enhances mid-range probability densities while suppressing boundary effects, leading to a self-regulating probability distribution that resists both excessive diffusion and over-concentration.

### 5.3. Spectral Representation

The Type-III derangetropy functional defines a structured probability transformation governed by a frequency-modulated mechanism. As a valid PDF, this transformation induces a well-defined characteristic function, encapsulating its spectral properties. The following theorem establishes the explicit form of the characteristic function associated with Type-III derangetropy.

**Theorem** **7.**
*The characteristic function of the distribution induced by the Type-III derangetropy transformation is given by*

(39)
$$\varphi_{\nu }\left(t\right)=\varphi_{0}\left(t\right)-\frac{1}{2}\left[\varphi_{F}^{+}\left(t\right)+\varphi_{F}^{-}\left(t\right)\right],$$

*where*

$$\varphi_{0}\left(t\right)=\int_{-\infty }^{\infty }e^{itx}f\left(x\right)dx,$$

*is the characteristic function of the original density, and*

$$\varphi_{F}^{+}\left(t\right)=\int_{-\infty }^{\infty }e^{itx}e^{i2\pi F(x)}f\left(x\right)dx,\;\varphi_{F}^{-}\left(t\right)=\int_{-\infty }^{\infty }e^{itx}e^{-i2\pi F(x)}f\left(x\right)dx,$$

*are modulated characteristic functions incorporating phase shifts.*


**Proof.** See Appendix B. □

Applying this result to the special case of a uniform distribution on (0,1), we obtain(40)$$\varphi_{\nu }\left(t\right)=\frac{e^{it}-1}{it}-\frac{1}{2}\left(\frac{e^{i(t+2\pi )}-1}{i(t+2\pi )}+\frac{e^{i(t-2\pi )}-1}{i(t-2\pi )}\right).$$ The characteristic function of the Type-III derangetropy functional for a uniform distribution follows a structured transformation process, which can be expressed in terms of the characteristic function of the base distribution. Given that the characteristic function of a uniform random variable is(41)$$\varphi_{0}\left(t\right)=\frac{e^{it}-1}{it},$$
the transformation induced by the Type-III derangetropy functional modifies the spectral representation as follows:(42)$$\varphi_{\nu }\left(t\right)=\varphi_{0}\left(t\right)-\frac{1}{2}\left[\varphi_{0}\left(t+2\pi \right)+\varphi_{0}\left(t-2\pi \right)\right].$$

This transformation defines an operator T that acts recursively on the characteristic function, introducing frequency shifts at ±2π and enforcing a structured refinement process in Fourier space. The recursive application of T induces a diffusion-like behavior, where successive iterations lead to a spectral stabilization process governed by a structured decay. The following theorem formalizes this structured diffusion process, showing that the characteristic function of the sequence of transformed distributions satisfies a recurrence relation.

**Theorem** **8.**
*Let $X_{n}$ be a sequence of random variables whose characteristic function evolves under the transformation*

(43)
$$\varphi_{n}\left(t\right)=T\left[\varphi_{n-1}\right]\left(t\right)=\varphi_{n-1}\left(t\right)-\frac{1}{2}\left[\varphi_{n-1}\left(t+2\pi \right)+\varphi_{n-1}\left(t-2\pi \right)\right],$$

*and define the re-normalized iterate*

$${\tilde{X}}_{n}:=\sqrt{2\pi^{2}n}\left(X_{n}-m\right)$$

*with corresponding ${\tilde{\varphi }}_{n}\left(t\right)=E\left[e^{it{\tilde{X}}_{n}}\right]$, where m denotes the median of the distribution. Then, the scaled sequence ${\tilde{X}}_{n}$ converges in distribution to a standard Gaussian; i.e.,*

(44)
$$\lim_{n\to \infty }{\tilde{\varphi }}_{n}\left(t\right)=e^{-t^{2}/2},\;\;\mathrm{for}\;\mathrm{all}\;t\in R,$$

*with convergence rate $O\left(\frac{1}{n}\right)$ in the variance collapse.*


**Proof.** We begin by analyzing the recursive evolution of the characteristic functions $\varphi_{n}\left(t\right)$ under the transformation(45)$$\varphi_{n+1}\left(t\right)=\varphi_{n}\left(t\right)-\frac{1}{2}\left[\varphi_{n}\left(t+h\right)+\varphi_{n}\left(t-h\right)\right],\;\mathrm{with}\;h=2\pi .$$ To approximate the right-hand side, we apply a second-order Taylor expansion of $\varphi_{n}$ around the point *t*, assuming sufficient smoothness, which yields(46)$$\varphi_{n}\left(t+h\right)+\varphi_{n}\left(t-h\right)=2\varphi_{n}\left(t\right)+h^{2}\varphi_{n}^{\prime \prime }\left(t\right)+O\left(h^{4}\right).$$ Substituting this expansion into the recurrence yields the following approximation(47)$$\varphi_{n+1}\left(t\right)=\varphi_{n}\left(t\right)-\pi^{2}\varphi_{n}^{\prime \prime }\left(t\right)+O\left(h^{4}\right),$$
which reveals that the recursive map approximately evolves according to a discrete second-order differential operator in Fourier space.Next, we consider the behavior of the variance of $X_{n}$. Since $\varphi_{n}\left(t\right)$ is the characteristic function of a random variable with median *m*, and assuming finite second moments, a second-order expansion around the origin gives(48)$$\varphi_{n}\left(t\right)=1-\frac{1}{2}\sigma_{n}^{2}t^{2}+o\left(t^{2}\right),\;\mathrm{as}\;t\to 0,$$
where $\sigma_{n}^{2}:=\mathrm{Var}\left(X_{n}\right)$. Substituting this expansion into the evolution equation for $\varphi_{n}$ leads to(49)$$\sigma_{n+1}^{2}=\sigma_{n}^{2}-2\pi^{2}+o\left(1\right),\;\mathrm{as}\;n\to \infty .$$ This recurrence implies that the variance decreases monotonically with *n*, and in particular, it decays according to $\sigma_{n}^{2}=O\left(1/n\right)$. Consequently, the sequence $X_{n}$ collapses in $L^{2}$ toward the constant value *m*; i.e.,(50)$$\sigma_{n}^{2}\to 0\;\mathrm{and}\;X_{n}\overset{L^{2}}{\to }m.$$We now analyze the asymptotic behavior of the re-scaled random variables ${\tilde{X}}_{n}:=\sqrt{2\pi^{2}n}\left(X_{n}-m\right).$ Let ${\tilde{\varphi }}_{n}\left(t\right):=E\left[e^{it{\tilde{X}}_{n}}\right]$ denote the characteristic function of ${\tilde{X}}_{n}$, and define the following auxiliary function(51)$$\psi_{n}\left(t\right):={\tilde{\varphi }}_{n}\left(\frac{t}{\sqrt{2\pi^{2}n}}\right).$$ This scaling is chosen so that $\psi_{n}\left(t\right)$ captures the low-frequency behavior of ${\tilde{\varphi }}_{n}$ near the origin, where Gaussian approximations are valid.Expanding ${\tilde{\varphi }}_{n}$ to second order, and using the earlier estimate $\mathrm{Var}\left(X_{n}\right)∼\frac{1}{2\pi^{2}n}$, we obtain the following recurrence(52)$$\psi_{n+1}\left(t\right)=\psi_{n}\left(t\right)\left(1-\frac{t^{2}}{2n}+o\left(\frac{1}{n}\right)\right).$$ Taking logarithms and summing over *k*, we obtain(53)$$\log\psi_{n}\left(t\right)=\sum_{k=1}^{n}\log\left(1-\frac{t^{2}}{2k}+o\left(\frac{1}{k}\right)\right)=-\frac{t^{2}}{2}\log n+o\left(\log n\right).$$ Exponentiating both sides yields(54)$$\psi_{n}\left(t\right)=e^{-\frac{1}{2}t^{2}+o\left(1\right)}⟶e^{-t^{2}/2},\;\mathrm{as}\;n\to \infty .$$ Therefore, the characteristic functions ${\tilde{\varphi }}_{n}\left(t\right)$ converge pointwise to the standard Gaussian characteristic function. By Lévy’s continuity theorem, this establishes(55)$${\tilde{X}}_{n}\overset{d}{\to }N\left(0,1\right),$$
as n→∞, which completes the proof. □

The Type-III derangetropy transformation defines a structured probability refinement process that induces an diffusion mechanism in Fourier space. The evolution of the characteristic function follows a discrete approximation to the heat equation(56)$$\frac{\partial \varphi_{n}}{\partial n}=-\pi^{2}\frac{\partial^{2}\varphi_{n}}{\partial t^{2}},$$
which describes how probability mass redistributes over successive iterations. Unlike uniform diffusion, which spreads probability evenly, Type-III derangetropy suppresses high-frequency oscillations while preserving structured probability redistribution, leading to a controlled refinement process. The sinusoidal modulation term $\sin^{2}\left(\pi F\left(x\right)\right)$ plays a central role in shaping this evolution. Rather than introducing oscillatory probability mass adjustments, this term guides structured refinement by selectively suppressing high-frequency variations, ensuring that probability mass does not dissipate arbitrarily.

The iterative application of Type-III derangetropy ultimately converges to a Gaussian distribution. This convergence is a direct consequence of the underlying diffusion mechanism in Fourier space, where the transformation systematically smooths the probability density while preserving structured patterns. The final equilibrium state corresponds to a probability density with maximal entropy under the imposed constraints, reinforcing the well-known result that repeated refinement of a distribution under structured diffusion leads to a Gaussian limit.

## 6. Applications and Examples

We present the application of Type-III derangetropy to quantum systems where probability amplitudes exhibit phase coherence and radial localization. A particularly instructive setting arises in the ground state of the hydrogen atom, governed by the time-independent Schrödinger equation in spherical coordinates. In atomic units ($\hbar =m=a_{0}=1$), the normalized spatial wavefunction for the 1s orbital is given by(57)$$\psi_{1s}\left(r,\theta ,\phi \right)=\frac{1}{\sqrt{\pi }}e^{-r}.$$ By integrating over the angular components, the corresponding radial probability density becomes(58)$$f\left(r\right)=|\psi_{1s}{\left(r\right)|}^{2}\cdot 4\pi r^{2}=4r^{2}e^{-2r},$$
which characterizes the likelihood of locating the electron at a radial distance *r*. The mode of this distribution is located at r=1, corresponding to the most probable electron–nucleus separation.

To apply the Type-III derangetropy transformation, we first evaluate the CDF as follows:(59)$$F\left(r\right)=\int_{0}^{r}4s^{2}e^{-2s}\;ds=1-e^{-2r}\left(1+2r+2r^{2}\right),$$
and substitute it into the transformation(60)$$\nu_{f}\left(r\right)=2\sin^{2}\left(\pi F\left(r\right)\right)f\left(r\right)=8r^{2}e^{-2r}\sin^{2}\left[\pi \left(1-e^{-2r}\left(1+2r+2r^{2}\right)\right)\right].$$ This yields a phase-modulated refinement of the original density that preserves normalization while amplifying regions near the cumulative midpoint F(r)=0.5.

To locate the mode of the transformed distribution, we solve the transcendental equation(61)$$F\left(r^{*}\right)=\frac{1}{2}\;⟺\;e^{-2r^{*}}\left(1+2r^{*}+2r^{*2}\right)=\frac{1}{2}.$$ Solving analytically gives(62)$$r^{*}=\frac{5-\sqrt{7}}{2}\approx 1.26.$$ This root identifies the radius at which the cumulative distribution reaches its midpoint. While the original mode at r=1 marks the most probable position, the shifted value $r^{*}\approx 1.26$ aligns with the median of the radial probability mass, emphasizing a globally balanced spatial configuration under sinusoidal modulation.

From a quantum mechanical standpoint, this median radius carries interpretive significance. It marks the location where half the electron’s total spatial probability lies within the interior sphere. In this context, the Type-III derangetropy amplifies regions of dynamic equilibrium in the electron cloud, those neither entirely confined near the nucleus nor completely delocalized. Such regions often play critical roles in measurement, transition probabilities, and interaction cross-sections.

Figure 3 displays the original radial density f(r) alongside the derangetropy-transformed function $\nu_{f}\left(r\right)$. The vertical dashed line indicates the shifted mode $r^{*}\approx 1.26a_{0}$. The result illustrates how Type-III derangetropy reweights the density to highlight the core quantum region associated with balanced probabilistic localization, offering a phase-aware alternative to conventional entropy-based measures.

## 7. Conclusions and Future Work

This work introduced a generalized framework for derangetropy functionals, information-theoretic operators designed to model the redistribution of probability mass in systems characterized by cyclical modulation, feedback dynamics, and structural fluctuation. By developing and analyzing three distinct classes of derangetropy transformations, Type-I, Type-II, and Type-III, we demonstrated how entropy modulation and phase sensitivity can be systematically encoded into functional mappings on probability densities. Each class corresponds to a qualitatively distinct mode of informational dynamics: entropy-attenuating sharpening, entropy-amplifying dispersion, and oscillatory redistribution.

Through formal derivations, we established that these functionals are governed by nonlinear differential equations whose behavior reflects the underlying informational geometry. In particular, we showed that derangetropy, when applied recursively, induces a diffusion process governed by the heat equation in the spectral domain, ultimately converging to a Gaussian characteristic function. This convergence theorem offers a unifying analytical perspective across all three classes and furnishes a tractable means to examine long-term behavior under cyclic modulation.

In addition to such theoretical directions, several application-oriented extensions remain to be pursued. One natural path involves developing multivariate analogues of derangetropy functionals. While the current framework operates on univariate distributions, many real-world systems, including multi-region neural circuits and high-dimensional ecological models, exhibit structured dependencies that require joint distributional analysis. Extending the derangetropy framework to accommodate tensor-valued or matrix-variate distributions may allow for modeling higher-order interactions and spatiotemporal coupling.

Another direction involves embedding derangetropy into learning architectures. For instance, entropy-modulated functionals can serve as non-parametric regularizers or adaptive filters in recurrent neural networks, where they may offer improvements in stability, sparsity, or interpretability. Likewise, incorporating derangetropy transformations into variational inference or generative modeling frameworks could enable new forms of distributional control grounded in cyclic or feedback-sensitive priors.

A promising direction for future research lies in studying its temporal behavior within time-evolving systems governed by Markov chains or stochastic differential equations. These analyses could elucidate how information evolves across state transitions in Markov processes or how it transforms across multiple timescales in stochastic systems.

Furthermore, while this study emphasizes the foundational development of the derangetropy framework, we acknowledge the importance of benchmarking it against other information-theoretic tools such as transfer entropy and directed information. Future research will involve comprehensive comparative studies to evaluate how derangetropy performs relative to existing frameworks in terms of localization, spectral sensitivity, and dynamic interpretability.

Collectively, the generalized derangetropy framework presented here offers a novel and analytically tractable approach to modeling structured information flow. By situating information redistribution within a family of entropy-sensitive and self-referential functionals, this work lays the foundation for future developments at the interface of information theory, dynamical systems, and probabilistic modeling.

## Figures and Tables

**Figure 1 entropy-27-00608-f001:**
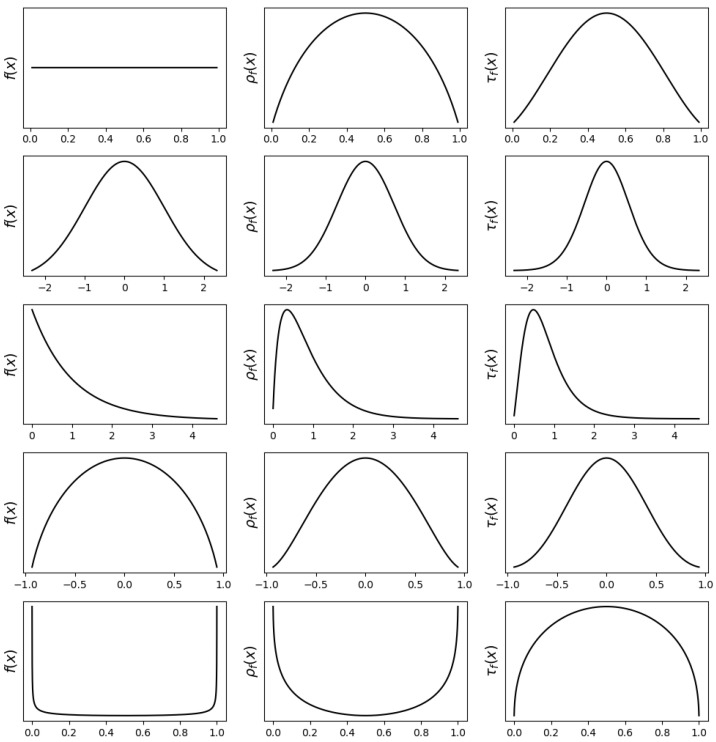
Plots of probability density functions f(x) (**left**), Type-I derangetropy functionals $\rho_{f}\left(x\right)$ (**middle**), and Type-II derangetropy functionals $\tau_{f}\left(x\right)$ (**right**) for uniform(0,1), normal(0,1), exponential(1), semicircle(−1,1), and arcsine(0,1) distributions.

**Figure 2 entropy-27-00608-f002:**
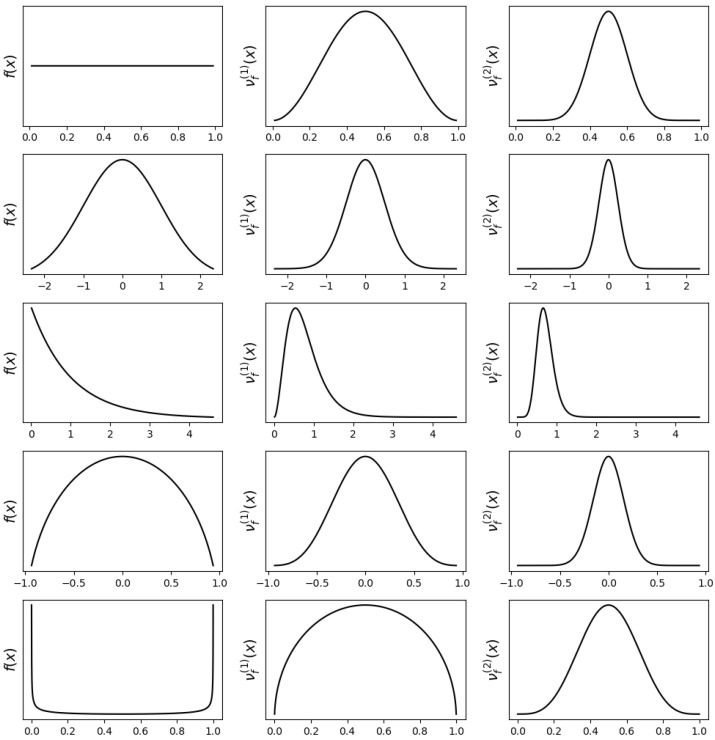
Plots of probability density functions f(x) (**left**), first-level Type-III derangetropy functionals $\nu_{f}^{\left(1\right)}\left(x\right)$ (**middle**), and second-level Type-III derangetropy functionals $\nu_{f}^{\left(2\right)}\left(x\right)$ (**right**) for uniform(0,1), normal(0,1), exponential(1), semicircle(−1,1), and arcsine(0,1) distributions.

**Figure 3 entropy-27-00608-f003:**
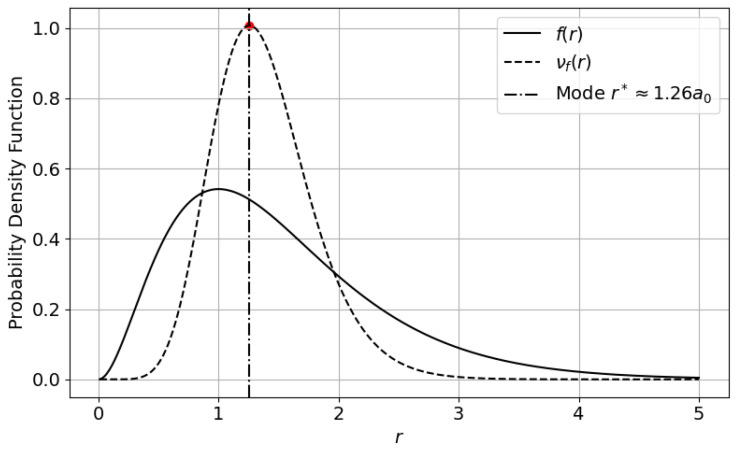
Type-III derangetropy applied to the radial probability density of the hydrogen atom’s ground state. The original density f(r) and the transformed density $\nu_{f}\left(r\right)$ are shown. The vertical dashed line marks the mode of the transformed distribution at $r^{*}\approx 1.26a_{0}$.

## Data Availability

No new data were created or analyzed in this study. Data sharing is not applicable to this article.

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
