# Peer review of "Generalized Derangetropy Functionals for Modeling Cyclical Information Flow"

_entropy, 2025, doi:10.3390/e27060608_

Round 1
Reviewer 1 Report
Comments and Suggestions for Authors
This paper introduces a framework for modeling cyclical and feedback-driven information flow through a generalized family of entropy-modulated transformations called derangetropy functionals. It is a valuable study. However, the quality of the paper can be improved as follows:
1- The "Related Work" (Literature Review) section is missing. A new section between "Introduction" and "Section 2" may be added.
2- "Type-III Derangetropy" can be explained in more detail.
3- Can some real-world application examples be given to indicate "how the proposed framework can be used in real-world systems?"
4- A table can be provided to summarize the related work to increase the understanding of the differences from the previous studies.
5- The paper contains many symbols. To increase readability, a notation table may be added to list all symbols and their meanings.
6- Some abbreviations are used in the paper without being defined, such as EEG and fMRI.
The authors should provide the full forms of these abbreviations upon their first use.
7- References are not sufficient. In the reference list, only four papers were published after 2020.
I suggest the authors cite the more recent papers (especially published after 2020).
8- The lack of comparison
Comparisons can increase the quality of the study.
Author Response
Please find the attached response letter.

Reviewer 2 Report
Comments and Suggestions for Authors
The authors study a new measure of entropy, which they call derangetropy, introducing different functional "types" of derangetropy that should be better suited for different use cases. The paper is, in general, relatively easy to follow and the presented results are clear. There are, however, issues with the general research line that I have failed to completely understand. I hope the authors can clarify them before the paper is published.
Major comments
a) The main difference between standard (Shannon or not) entropy and all types of deragentropy that the authors explore is that, while entropy measures are defined as scalar values assigned to entire probability distributions, deragentropy is instead a function of the random variable (see eg definition 1). Not only this, derangetropy is itself normalized in the domain of the original probability distribution, and thus it is itself a pdf. This is a rather strong departure from the standard interpretation of entropy. I am not sure I fully understand the rationale behind this choice, and how derangetropy is related to entropy at all.
b) Related to the previous point, Shannon entropy is typically interpreted as a measure of uncertainty. Is the derangetropy distribution describing uncertainty? If so, why?
c) The authors refer multiple times to "information flow". However, they do not seem to refer to standard measures of information flow such as transfer entropy or mutual information rates. What are they referring to exactly? See reference [9] for instance.
d) The authors briefly mention mutual information in the introduction. Does the derangetropy framework have an analogous of mutual information?
e) The idea that derangetropy can be applied to study information dynamics is interesting, but dynamics is not studied in detail. Could the authors study the evolution of derangetropy for any stochastic process? For instance, the Gaussian propagator of an Ornstein-Uhlenbeck process is known analytically. What kind of propagator does it induce for the derangetropy distribution? What role do timescales play (see for instance Physical Review X 14 (2), 021007 2024 for the effect of timescales in mutual information)?
f) Is there an equivalent of derangetropy for discrete distributions?
Minor comments
a) My understanding is that the authors first introduced deragentropy in their previous work ([2] in the paper). It was not immediately clear to me that this is a relatively new and unexplored framework that the authors introduced, and it would be better in my opinion to state this clearly in the introduction.
b) The functional forms of the different derangetropy types are highly nontrivial. Can they be derived from first principles? Are there any other functional forms that perform equally well or better?
c) The references are highly selective and do not include many concrete examples. I think the authors, if they want, could expand them by including more references (see for instance the recent review by Tkacik and ten Wolde, "Information Processing in Biochemical Networks", Annual Review of Biophysics, 2025 or the more classic one, "Information processing in living systems" by Tkacik and Bialek, Annual Review of Condensed Matter Physics 2016) and relate future applications to physical and biological systems where derangetropy could make useful predictions.
Author Response
Please see the attached response letter.

Round 2
Reviewer 1 Report
Comments and Suggestions for Authors
The authors revised the manuscript adequately according to the reviewer comments.
The manuscript is now more qualified and clear.
I have no further comments.
I suggest accepting it for publication in its present form.
Reviewer 2 Report
Comments and Suggestions for Authors
I thank the authors for replying to my comments, which clarified most of my doubts. The manuscript can now be accepted, in my opinion.
Nevertheless, I still have some minor doubts about the interpretation of derangetropy in terms of information. The reason why information theory is based upon quantities such as the Shannon entropy or, more in general, statistical divergences, is that there is a deep connection to the geometry of the statistical manifold where the probability distributions at hand are defined. This provides a quantitative interpretation of these quantities. If I may, I would like to suggest that the authors try to understand if such connections exist for derangetropy as well. I understand that this question is well beyond the scope of the present manuscript, but it may be of interest for future studies.